# Influence of Neighborhood Size and Cross-Correlation Peak-Fitting Method on Location Accuracy [note 1]

**DOI:** 10.3390/s20226596

**Published:** 2020-11-18

**Authors:** María-Baralida Tomás, Belén Ferrer, David Mas

**Affiliations:** University Institute of Physics Applied to the Sciences and Technologies, University of Alicante, P.O. Box 99, 03080 Alicante, Spain; maria.baralida@ua.es (M.-B.T.); belen.ferrer@ua.es (B.F.)

**Keywords:** peak-locking, cross-correlation, subpixel, Gaussian fitting, thin-plate splines, polynomial fitting

## Abstract

A known technique to obtain subpixel resolution by using object tracking through cross-correlation consists of interpolating the obtained correlation function and then refining peak location. Although the technique provides accurate results, peak location is usually biased toward the closest integer coordinate. This effect is known as the peak-locking error and it strongly limits this calculation technique’s experimental accuracy. This error may differ depending on the scene and algorithm used to fit and interpolate the correlation peak, but in general, it may be attributed to a sampling problem and the presence of aliasing. Many studies in the literature analyze this effect in the Fourier domain. Here, we propose an alternative analysis on the spatial domain. According to our interpretation, the peak-locking error may be produced by a non-symmetrical sample distribution, thus provoking a bias in the result. According to this, the peak interpolant function, the size of the local domain and low-pass filters play a relevant role in diminishing the error. Our study explores these effects on different samples taken from the DIC Challenge database, and the results show that, in general, peak fitting with a Gaussian function on a relatively large domain provides the most accurate results.

## 1. Introduction

Cross-correlation is a useful technique for establishing similarity between two signals. As correlation can derive from minimizing the mean square error between two signals or images [1], it is a robust tool for comparing images corrupted by Gaussian noise, which is the normal case in most circumstances during image processing where illumination is good enough. Apart from giving a similarity metric, peak location describes the position where the reference image and template match present maximum coincidence and is, thus, often used for image aligning or object tracking in a scene.

Despite the many advantages and applications of the cross-correlation [2,3,4,5,6], its standard formulation presents two main drawbacks: the dependence of the correlation result on image and template amplitudes to, thus, produce high peaks when a dark template is compared to a bright object or vice versa, even though their similarity is minimum [7] and the limited resolution, which is set, by construction, to one pixel.

The first issue is solved by using a normalized cross-correlation algorithm, which is the common approach when dealing with images. Regarding the second issue, subpixel resolution can be achieved by interpolation, which can be applied to either image before calculating their cross-correlation [8], the correlation function itself, to increase the accuracy in the location of its maximum [9]. The first approach is frequently followed to analyze deformations in solid materials as it allows for deformation mappings to be easily implemented [8]. The second one, i.e., peak interpolation, has faster and easier applications in non-deforming scenes and is, thus, adequate for aligning and tracking isolated objects. In our case, we pay attention to the second technique.

Briefly, the technique consists of interpolating the correlation function over a local area around the maximum peak and then refining the search by fitting the peak neighborhood to an analytical function [8]. This procedure may increase peak location accuracy by almost two orders of magnitude [10,11]. Despite the evident improvement, the procedure also introduces a bias error, which limits its performance. The error, known as the peak-locking or pixel-locking effect, means that the peak location obtained through local fitting is always biased toward the closest integer coordinate [12].

The origin of peak-locking has been usually attributed to an aliasing effect due to a poor image texture [13] combined with an inadequate choice of the interpolant function [14]. The problem of the aliasing can be avoided with adequate sensors and lenses. In general, according to the Nyquist limit, a pseudospeckle scene will be well sampled when the dot unit is larger than 2 px [15].

Properly choosing the interpolating function is a more delicate issue and can be better explained in the spatial domain. Consider a scene with a low-noise object and a template containing a shifted version of the object. The finer the details of the object, the narrower the correlation peak will be, since a small displacement will degrade the correspondence between object and scene [15]. If one considers a narrow local domain around the maximum of the correlation function, there will be a reduced number of samples to fit to the interpolant function. Therefore, this maximum may have an excessive weight in the fitting, and it may pull the recalculated maximum to its position in the original grid. Consequently, a bias area towards the nearer integer corresponding to the original maximum position is introduced. Notice that when the maximum is exactly in the middle of two pixels the weight distribution is balanced, and the error is 0.

One can take a larger neighborhood and thus a larger number of samples to fit in order to compensate for the excessive weight of the correlation maximum, but this would eventually include information of non-correlated positions and thus distort the final result. Additionally, since the number of samples is still not very large, any secondary peak in the neighborhood will also unbalance the fitting and pull the fitted maximum towards it.

According to this, fitting functions that only consider the peak area, i.e., quadratic functions, may show a good fitting with a small neighborhood, but would be affected by the peak-locking effect. An extensive function that considers the peak and the region around, i.e., Gaussian fitting, would compensate for this effect but would need a larger neighborhood. Therefore, a proper selection of the interpolant function together with the interpolation domain is critical to decrease the peak-locking error.

In [16], the authors proposed different interpolation algorithms in an 8×8 neighborhood, and showed that the Gaussian function provided better results than the bilinear one, third-order polynomial or bicubic splines.

An additional strategy consists of filtering out the finer details of the sequence being analyzed, so that the correlation peak is softer and thus more suitable to a fitting by an analytical and derivable function. In [17], the authors introduced a defocus into the image-capturing process with an effect of reducing the peak-locking effect. This blurring was introduced experimentally by manipulating the objective. By doing so, and by adjusting the correlation peak through a Gaussian function, the results were slightly improved compared with the non-filtered results. This effect was thoroughly analyzed in [18]. Michaelis et al. [19] tested a different configuration for blurring an image using an optical diffuser. They also implemented two different interpolation functions (splines and bicubic), which gave good results for very small particle sizes.

Another strategy to diminish the peak error consists of maximizing the dynamic range of the image and template [20]. It has been shown that the accuracy in object-tracking tasks is directly related to the number of gray levels of the image [10,19]. Nevertheless, images and their luminance dynamic range are linked to the experimental setup. Hence, albeit important, it is not often a parameter that we can modify at will.

In this manuscript, we propose a combined analysis of three factors that may help to compensate for the peak-locking error and increase the accuracy of the proposed methods. Therefore, we will analyze the interaction between the two mentioned fitting functions, Gaussian and quadratic, in order to analyze their dependence with the neighborhood and its capability to reduce the error. The analysis will be complemented with the analysis of the results obtained using spline fitting. These fitting functions are more adaptable than the other two, so they may be useful in a wide range of neighborhood sizes. Additionally, the effect of a Gaussian defocus on the scene and the template is also analyzed together with the interaction with the fitting function.

The final aim of this paper is to determine which is the best method, which includes fitting function, application domain and amount of defocus, for obtaining object displacement with reduced peak-locking error. Due to the large amount of variability in images, we took a set of speckle sequences from the 2D-DIC public images bank from the Society for Experimental Mechanics (SEM) [21]. We selected the first image as a reference and the texture on it was tracked through the sequence.

Preliminary results of this analysis were presented at the SPIE Photonics Meeting 2020 [22], concluding that quadratic functions are more suitable for small neighborhoods, while Gaussian functions stand for large neighborhoods without further analysis. In what follows, we will explain the reasons for that and a final rule of thumb.

## 2. Methods

The purpose of the simulation is to analyze the relationship between three different peak-fitting methods and the neighborhood size together with the influence of the low-pass filters on the peak-locking method. In order to make our conclusions more general and facilitate the reproducibility of our results, we have checked our method with synthetical images taken from an image bank. Therefore, the variability due to the setup, or noise in the image, is excluded.

We tested five different sequences taken from the 2D-DIC image bank provided by the Society for Experimental Dynamics [21]. These sequences, which were synthetically generated, come with a full description of the movement and subpixel displacement of the texture, and are often used as testing benches for tracking algorithms. The sequences from 2D-DIC are images of random dots whose contrast, noise and shift differ in the distinct sequences. Figure 1 depicts the selected samples and their properties. Sequences contain horizontally and vertically shifted versions of the first image according to the specified steps. The number of frames in each sequence is the amount needed to accomplish a one-pixel accumulated displacement.

From each sequence, the provided reference was selected as the template and its position was tracked throughout the sequence by using the cross-correlation operation which is implemented here through the normalized cross-correlation algorithm, *normxcorr2*, in Matlab [23]:(1)$$\gamma \left(u,v\right)=\frac{\sum_{x,y}\left[f\left(x,y\right)-\bar{f_{u,v}}\;\right]\;\left[t\left(x-u,y-v\right)-\bar{t}\;\right]}{\sqrt{\sum_{x,y}{\left[f\left(x,y\right)-\bar{f_{u,v}}\;\right]}^{2}\sum_{x,y}{\left[t\left(x-u,y-v\right)-\bar{t}\;\right]}^{2}\;}},$$
where *f* is the image taken as a reference and *t* the template,$\;\bar{t}$ is the mean value of the template and $\bar{f}$ is the mean value of *f(x,y)* in the region under the template.

The test was carried out at full field, i.e., taking all the image as the template. Nevertheless, a frame of 8 pixels on all sides was imposed on the template in order to prevent the shifted image moving outside the boundaries of the reference image [24].

After obtaining the correlation function, a small region around the peak is selected and fitted to a soft function. These operations eventually relocate the peak inside a pixel region so that its maximum can be recalculated with incremented accuracy. As stated in the Introduction, we use a Gaussian function, a second-order polynomial and cubic spline [19,25] as fitting functions on different neighborhood sizes around the correlation peak. Cubic splines have been implemented through the thin-plate spline algorithm provided by Matlab, which provides a smoother fitting function [26,27].

The fitting was calculated on different neighborhood areas around the maximum of the correlation peak (Nbd). This size was taken from 3×3 to 11×11 pixels, with the peak centered in the region, so only odd sizes were considered. Average speckle size in the samples used was estimated through autocorrelation, showing an average radius larger than 5 pixels. Therefore, areas larger than 11×11 would include unmatched results, which may distort the error estimation. We also tested the influence of defocusing on the accuracy of the tracking results. In mathematical terms, blurring can be described by a convolution (3) of the image with a Gaussian function (2):(2)$$G_{r_{b}}\left(x,y\right)=exp\left(-\frac{x^{2}+y^{2}}{2\cdot r_{b}^{2}}\right),$$
(3)$$f_{r_{b}}\left(x,y\right)=f⨀G_{r_{b}}\left(x,y\right)=\underset{u,v}{\sum }f\left(u,v\right)G_{r_{b}}\left(x-u,y-v\right),$$
where $r_{b}$ is the blur radius.

In principle, if the camera is defocused throughout the capturing process, both the reference image and template will be blurred, so the cross-correlation between the blurred image and template can be written as (4):(4)$$C_{r_{b}}=f_{r_{b}}⨂t_{r_{b}}=\underset{u,v}{\sum }f\left(u,v\right)G_{r_{b}}\left(x+u,y+v\right),$$
where, for simplicity’s sake, we used the general definition of correlation instead of the normalized one. In any case, generalization is straightforward.

According to the basic properties of both correlation and convolution, we can rewrite Equations (4) as (5):(5)$$C_{r_{b}}=f_{r_{b}}⨂t_{r_{b}}=f_{r_{b}}⨂\left(t⨀G_{r_{b}}\right)=\left(f_{r_{b}}⨂t\right)⨀G_{r_{b}}=\left[\left(f⨀G_{r_{b}}\right)⨂t\right]⨀G_{r_{b}},$$

As the Gaussian function is symmetric, the above-written expression can be finally expressed as (6):(6)$$C_{r_{b}}=f_{r_{b}}⨂t_{r_{b}}=\left(f_{r_{b}}⨂t\right)⨀G_{r_{b}}=\left[\left(f⨂t\right)⨀G_{r_{b}}\right]⨀G_{r_{b}},$$

Thus, we can see that the effect of the blurred reference and template is a double blurring of the correlation peak. As blurring was symmetrical, the main effect was to soften the peak to, thus, make a more adequate profile for accurate fitting. Unfortunately, a double defocus can introduce excess blurring, and can also degrade the function and mask the peak, which would cancel out the obtained advantages. Therefore, it is worth analyzing the amount of blurring that provides the best possible results. Accordingly, an analysis to compare a sharp reference with a blurred template was done and is presented in the Discussion in order to check whether the double-blurring filter was redundant or not.

The calculation process started by introducing a Gaussian filter with radius $r_{b}$ to both the image and template before calculating the normalized cross-correlation. The value of the radius was varied from $r_{b}=0$ (delta function, no blur) to $r_{b}=5$. Figure 2 shows a flow chart with the algorithm implemented in Matlab. The background of the program is depicted in gray, whereas the specific parts of the fitting algorithms are represented in green (Gaussian fit), yellow (thin-plate splines) and pink (second-order polynomial fit) as they are shown in the Results. The depicted sequence was repeated for each sequence with a different $r_{b}$.

Briefly, each frame was compared with the first one in the sequence and the correlation function was obtained. Then a region around the correlation peak was fitted to three different functions over distinct neighborhood areas around the peak. The new peak position of the fitted function was then obtained. The new maximum was obtained through a minimum search of the inverse of the fitted functions, following the algorithm developed in [28]. Although it can be analytically calculated for the Gaussian and the quadratic case, we preferred to use the same method for all the functions in order to avoid distortions introduced by the calculation algorithms.

As the real movement was provided by the 2D-DIC image bank, our results were compared to the theoretical displacement to evaluate the errors and to obtain the best combination to measure target movement. The error was calculated through the mean error (μ) with its standard deviation (STD) and the maximum error (MaxErr) of each sequence:(7)$$\mu =\frac{\sum_{i=1}^{N}\left(x_{i_{calc}}-x_{i_{teor}}\right)}{N},$$
(8)$$STD=\sqrt{\frac{\sum_{i=1}^{N}{\left(\left(x_{i_{calc}}-x_{i_{teor}}\right)-\mu \right)}^{2}}{N}},$$
(9)$$MaxErr=\max\left(\left|x_{i_{calc}}-x_{i_{teor}}\right|\right).$$
where $x_{i_{teor}}$ is the reference value provided by the DIC Challenge site, $x_{i_{calc}}$ is the value calculated through the different methods and *N* is the number of samples. In our case, as the measurement extended throughout the sequence, *N* refers to the number of frames in the evaluated sequence.

The main programs, subroutines for calculation of the maxima and obtained results can be downloaded from [29] as Appendix A.

## 3. Results

We obtained the parameters in Equations (7)–(9) for all five samples with the three peak interpolation methods applied to neighborhoods of all odd sizes ranging from 3×3 to 11×11, and with six different Gaussian filters with radii ranging from 0 (no blur) to 5, which totaled 90 tests per sample. The obtained displacement values were compared to the data provided by the DIC database and the error was evaluated. As presenting all the results would be extensive, we selected the results according to the maximum error values by simply selecting the best and worst cases for each sample and interpolation method, which corresponded to the minimum and maximum MaxError, respectively. The results are summarized in Table 1. Note that, for Table 1, for remaining calculations and graphs presented in the manuscript, errors only refer to vertical shifts. The errors obtained from the horizontal displacements were similar, but their analysis was omitted to avoid a redundant analysis.

The results showed that the best results for all the methods had mean errors below 0.005 px and standard deviations below 0.006 px. Additionally, note that the maximum error (MaxErr) for the best result in each fitting function, which can be taken as a measure of the peak-locking effect, was at least one order of magnitude smaller than the sample shift. This means that the tracking of samples was very good, provided that the parameters were well selected.

The results presented in Table 1 can serve to set the extreme results that were obtained through the different methods herein presented but did not indicate the influence of the different parameters. In order to better understand the influence of the different parameters, we depicted the error for the three fitting functions for a fixed defocus parameter $r_{b}=2$ with all the possible neighborhood sizes in Figure 3. As the 3×3 neighborhood provided such bad results by the Gaussian and spline-fitting methods (see Table 1), the corresponding results were
deleted from the graph. The lines for samples 3 to 5 corresponding to the 5×5 area of the Gaussian fit were also deleted for the same reason. Note that a complete figure was added in the Matlab format as a Appendix A.

The first noticeable fact in the graphs is that not all the error curves presented the typical sigmoid shape with symmetry around the 0.5 pixel shift value due to the pixel-locking effect (see the errors for Sample 1 in Figure 3). This is especially noticeable in the Gaussian case, where three of the five samples do not even show any clear trend. This unusual behavior does not imply large errors since the graphs depicted for the Gaussian case are of the same order or lower than the error obtained by the other fitting functions. Moreover, for this fitting function, and except for the non-depicted cases, the error does not strongly depend on neighborhood size, provided that it is large enough.

In the splines case, the typical sigmoid shape only appears in Samples 1, 3 and 5. The curve for Nbd=5×5 presents a very different behavior to that of the other curves. Once again, if we do not consider the anomalous cases, i.e., Nbd=3×3 and Nbd=5×5, there seems no systematic error dependence on neighborhood size: although we can see some differences in each individual sample, it is relatively small and dependence on size is not the same in all cases.

Finally, the curves representing the errors calculated with the second-order polynomials present the typical shape due to the peak-locking effect. In this case, the error is bigger than that obtained with the other two functions and is of the same order in all the samples. Unlike what happened with the other two fitting functions, here, we notice a marked dependence of neighborhood size, where the bigger the error, the larger the interpolation area.

Figure 4 depicts our analysis of the influence of defocusing on the error, along with the error for fixed neighborhood Nbd=7×7 and changing defocus parameter $r_{b}$. At first glance, the curves share some similarities with the curves in Figure 3, i.e., lack of the typical sigmoid shape in the same cases as before. It is also noticeable that, in all those cases, there was no significant dependence on the defocusing parameter. Notwithstanding, and as before, the error obtained by fitting the peak with a Gaussian function is equal to or lower than the error obtained by other fitting functions.

Regarding the influence of the defocus, note that the more marked the defocus, the lower the error for all the cases calculated by the quadratic polynomial. This can also be stated for the spline method, although in this case, the larger difference lies between $r_{b}=0$ and all the other cases.

According to the depicted figures, it would seem that the results obtained using the Gaussian function for fitting the peak were independent of neighborhood size and image blurring, provided that the calculation area was large enough. Moreover, the results obtained by this method did not present the typical peak-locking shape, and errors were similar to or lower than in the other methods.

Despite the obtained results, we have only analyzed the error due to the calculation area for one fixed defocusing filter and the effect of the defocus for a fixed neighborhood, respectively. Therefore, in order to gain more insight into the error dependence on the analyzed parameters, Figure 5, Figure 6 and Figure 7 reveal the plots of the variation of the three error parameters for the row shifting expressed in Equations (7)–(9) (mean error, standard deviation and maximum error, respectively) with the defocus parameter for all the neighborhood sizes and for the three fitting functions herein analyzed.

Figure 5 illustrates the mean value of the error obtained for each case according to the radius of the blurring Gaussian filter. As we were interested only in the error magnitude, the absolute value of the error is represented. A different line is depicted for each neighborhood size. As we can see in Table 1, the Gaussian function does not provide good results with small interpolation areas. Therefore, the 3×3 neighborhoods graphs were deleted to facilitate the visualization of the other curves because of their large errors, with values higher than 0.5 px (see Table 1). For the same reason, the 5×5 neighborhood error curve obtained for the Gaussian function was also deleted for Sample 5, with a peak close to 0.1 px. Apart from the reported cases, the mean value of the error was below 0.05 px in most cases, which went below the imposed shift between frames. The complete figure was added as Appendix A to better allow the interpretation of the results.

At first glance, it would seem that the defocus increased the error when Gaussian or spline functions were used. In both cases, a defocus with radius $r_{b}=4$ seemed to produce an error reduction in some samples, but we hypothesize that this happened because the particular texture of this sequence and is not a general rule. Nevertheless, we can see that the error change due to the blurring filter was less than 1% of the shift (0.05 px in the first two samples and 0.1 in the other three). So the effect on the mean value could not be considered very strong, but could be important when using the quadratic fitting function with large interpolation areas.

Regarding the Nbd parameter, we can see that, for the depicted Gaussian and polynomial cases, and for Nbd=7×7, 9×9 and 11×11, the larger the interpolation area, the bigger the error, but not in all cases. Note also that the Nbd=3×3 case was somewhat anomalous because in the Gaussian and spline functions, the errors obtained for the smaller case were huge in all the samples. The 5×5 domain also produced large errors in Samples 3 to 5 with the Gaussian function and erratic behavior with splines. Finally, when quadratic functions are used, these two particular domains seemed to provide opposite results according to the general trend described for this case.

Despite this analysis, the mean value was a poor parameter for measuring the error. It described trends in the result but, as these sequences were artificially generated, no a strong bias was expected here, as previously seen in Figure 3 and Figure 4. In any case, we discovered that using very small interpolation areas may be problematic in the majority of cases.

Figure 6 offers the graph for the standard deviation (STD) for all the discussed cases. This parameter indicated the variability of the results. As in the previous case, the graphs with the highest values were deleted. This happened for all the curves corresponding to Nbd=3×3 in the Gaussian and spline cases, and to the 5×5 curves for the Gaussian case and Samples 3 to 5. Once again, the complete graph is included as Appendix A.

In the graphs, we can see that blurring may help to narrow the variability in the results. The strength of this effect depends very much on the fitting function. In the Gaussian case, the effect strongly depends on the sample as the benefit is observable only in Sample 1, while the error increases with blurring for the other samples. In the polynomial case, the effect is general, with better results for large $r_{b}$. For the spline method, the improvement is also noticeable, albeit very weak. Note that the effect on some samples is nonexistent, or even negative.

Regarding the influence of the neighborhood size, once again, Nbd=3×3 combined with Gaussian or spline functions resulted in wide variability and a large error, as mentioned above. For the Gaussian case, this dependence strongly depended on the sample as we observed the opposite behavior in different samples. With the polynomic function, the results were less dispersed the smaller the area was, which agrees with what is deduced from the table, but is the opposite to what happened with the mean error. However, the relation was very weak in that case. When spline fitting was applied, neighborhood size displays no clear dependence, except for the 3×3 case.

When considering the absolute value of the standard deviation, we find that, accordingly with the previous results, the Gaussian function generally gives lower values than the other two methods.

Another useful parameter for determining the performance of each fitting function is the maximum error, which may correspond to the peak of the peak-locking error. Figure 7 displays the graphs with the maximum error for each sequence, fitting method and neighborhood in front of the blurring radius. As in previous cases, the curves corresponding to the Gaussian and spline fitting methods in a 3×3 region were deleted from all the samples. The 5×5 regions in Samples 3 to 5 was also deleted for the Gaussian method as they gave values around 1 px which would not, thus, allow the other cases to be visualized.

We can see that the curves depicted for the maximum error are similar to those with the standard deviation, which implies that the maximum value of the peak-locking error is probably the main source of the errors in the calculation. Hence, the depicted results confirmed the conclusions drawn from the other graphs and no further comments will be added.

## 4. Discussion

The results depicted above confirm the hypotheses posed in the Introduction: quadratic functions provide good results for peak fitting, but are prone to peak-locking error, while Gaussian functions give more robust results, provided that the fitting area is large enough (see Figure 4 and Figure 7). As we said, narrow peaks are supported by a few samples, so the maximum weight determines the fitting result, pulling the recalculated maximum location towards the location of the sample where the original maximum is located. Including more samples would compensate the result but would also include areas outside the peak. Eventually, the fitting neighborhood would include the peak skirt and curvature changes which would no longer fit to a paraboloidal surface, and thus the error will increase.

Gaussian functions are capable of reproducing both the peak and the planar surrounding area, although more samples are required. Because of this, the weight of the maximum is compensated, and the location error is less prone to being affected by peak-locking error. Among both situations, spline functions provide reasonable results in all neighborhood sizes, but because of their adaptability, the fitting is also biased towards the peak maximum and, thus, it is also affected by the peak-locking effect.

In order to illustrate the adaptability of each fitting function to the peak surface, we have represented in Figure 8 the correlation peak corresponding to a shift of 0.4 px in the two extreme cases calculated for Sample 1; i.e., with 3×3 and 11×11 neighborhoods and no blur. Although the curves may have wide variability from frame to frame and for different samples, they serve to illustrate the point herein explained. We notice there that the Gaussian function cannot adapt to the skewed shape of the local area around the maximum, while the polynomic function correctly fits to it. In the case of the larger neighborhood, the Gaussian function can reproduce the curvature change while the quadratic function just reproduces a paraboloidal dome. As we said, the spline function can adapt to both situations.

The results here shown also explain the reason why the Gaussian function seems to be more insensitive to the neighborhood size once it is large enough (see Figure 3). Because of its particular shape, the weight of the samples farthest from the center is very low and therefore will not affect the location results. In the case of the spline function, there is not such distance compensation and the result may be affected. Finally, in the quadratic case, it is clear that larger neighborhoods produce larger errors.

Regarding the blurring filter, it has a double effect in the correlation surface. On the one hand, the peak gets softer, so the fitting by smooth functions is more accurate. This affects more noticeably the results obtained with the quadratic function which are very dependent on the central samples (see Figure 4 and Figure 7). On the other hand, blurring also decreases the weight of the eccentric samples, increasing the relative weight of the central sample and thus may slightly increment the location error obtained through Gaussian fitting. In any case, the effect introduced by the Gaussian filter is similar to that produced by the Gaussian fitting function, hence explaining why the error obtained through this fitting method is not very much affected by blurring or even increases in some cases. At this point, we wish to recall Equation (6) where the mathematical formulation of the blurring filter is explained. By comparing a blurred reference to a blurred template, we obtained a double-blurring of the correlation function. As this double application of the blurring filter is redundant, we hypothesized that it is possible to compare a sharp reference to a blurred template (or vice versa) without increasing errors. Thus, we tested the results when a sharp reference was compared to a blurred template. In Figure 9, we plotted the curves with the maximum errors.

As we can see, the results are similar to those obtained using a double defocus. This means that blurring is not a decisive parameter in the peak-locking error. Although it can help to improve the results, adding a Gaussian filter (or an experimental defocus) in both the images used for the normalized cross-correlation was redundant.

This “invariability” in the template defocus proved most convenient for some experimental implementations: long-time experiments may cause small mechanical drifts in the camera or the sample and, therefore, some frames may appear slightly blurred. A defocus may take place during experiments using short depth-of-field lenses and samples whose size may change due to heat dilation, tension or swelling [30]. The herein shown results demonstrate that these changes applied to one of the images being compared had no marked effect on the final result, which remained valid and, thus, indicated that the subpixel tracking through local interpolation was a robust method.

According to the results herein presented, we can reach several conclusions. Regarding the fitting function, we found that, except for the smallest fitting region, the Gaussian fit gave the smallest errors that are, in some cases, almost one order of magnitude smaller than the errors introduced by the other methods. However, the best results in all the functions were similar, which means that all the fitting functions would display similar performance under optimal conditions.

When Gaussian or spline functions are used, small areas around the peak should be avoided. With these two functions, and except for areas 3×3 and 5×5, the errors did not seem to depend on either neighborhood size or the blurring filter radius. This result agrees with the partial results represented in Figure 3 and Figure 4. On the contrary, when employing quadratic polynomials as fitting functions, both the neighborhood size and blurring strongly impacted the results, and the smaller the errors, the smaller the fitting area and the larger the defocus parameter.

The effect of blurring to improve the results was noticeable in many cases, but there are many exceptions. Hence, we cannot state that the use of blurring to diminish the error is a general benefit because it depends on both the fitting function and the sample. In any case, except for the Gaussian fitting functions, a minor defocus could help to narrow the variability of the results (taken as the standard error) and to slightly reduce the peak error. So, introducing it could be advisable.

Thus, in summary, we found the smallest error with a Gaussian fitting applied in a large neighborhood around the peak and with no blurring.

The results here obtained agree with those that appear in the literature [16], where the authors obtained better results for the Gaussian fitting than for the other methods by using an 8×8 neighborhood. In [17,19], the authors reported a minor improvement when using a slight defocus and a Gaussian fitting.

## 5. Conclusions

In this manuscript, we tested the accuracy of the commonest subpixel tracking methods based on cross-correlation. We focused on the methods that employ local interpolation in a small area around the correlation peak to refine the maximum location. To this end, we tested the influence of neighborhood size around the peak with three different fitting functions: Gaussian, thin-plate splines and second-order polynomials. We also checked the use of defocus as a strategy for diminishing the peak-locking error and how it was affected when that defocus changed along the image sequence. All the tests were carried out in five sequences taken from the DIC Challenge site [21].

We generally noticed that the three functions provided good accuracy, and slight blurring helped to increase accuracy, despite us finding slight variation among samples. The fitting functions provided different results depending on neighborhood size. Therefore, the Gaussian function provided the best results with large neighborhoods (11×11), while the second-order polynomial seemed to work better with small areas (3×3). The thin-plate function apparently worked correctly with any neighborhood size. Our tests reveal that the best result was obtained for the Gaussian function with a neighborhood of 11×11 and no defocus. However, as the worst adjustment was also achieved with the Gaussian function, it is important to correctly select the values of both the focus and the neighborhood depending on the fitting function. Blurring significantly improved the error for a second-order polynomial fitting, but had no clear trend for the other two studied functions.

Additionally, in the Methods section, we show that comparing two blurred images by a cross-correlation operation is the equivalent to comparing two sharp images and then introducing double deblurring into the correlation function. Accordingly, we recalculated the errors by comparing a sharp reference with a blurred template. The results showed that the performance of the three methods with all the different parameters were similar to the double deblurring case. So we can conclude that the methods presented herein are blur invariant, which means that the subpixel technique based on the interpolation of the correlation peak is robust for experimental implementations in which the focus may change due to drifts in the optical system or to the sample’s position change.

As a consequence of this, the rule of thumb that can be derived from our tests is that Gaussian fitting applied on large neighborhoods around the maximum of the correlation function may provide the most accurate results in pseudospeckle images without the need for blurring filters.

## Figures and Tables

**Figure 1 sensors-20-06596-f001:**
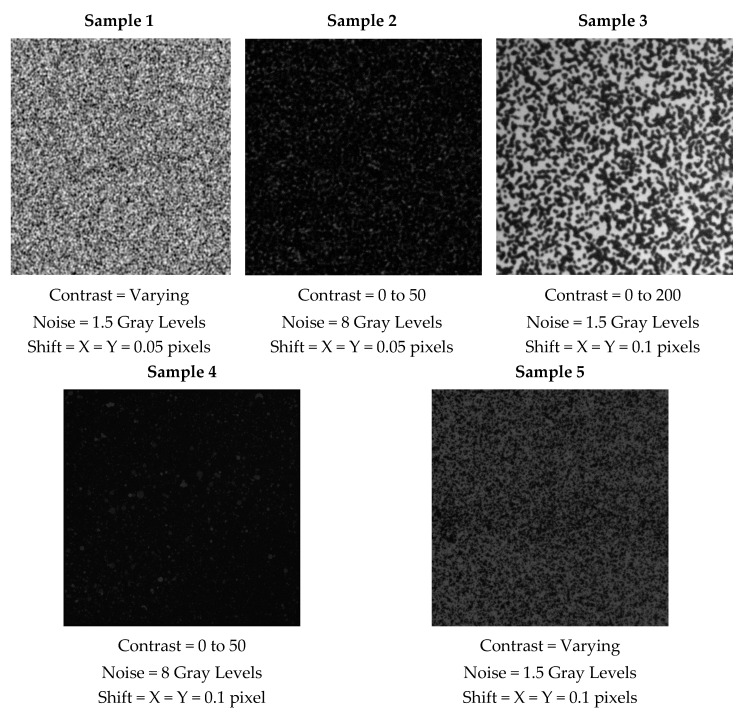
Images of each sample selected from 2-D DIC from the Society for Experimental Mechanics (SEM) with random dots and their properties [22].

**Figure 2 sensors-20-06596-f002:**
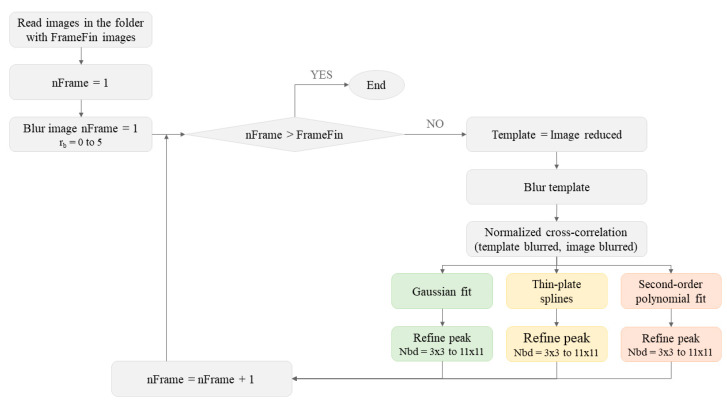
Flow chart with the algorithm implemented in Matlab to reduce the peak-locking error.

**Figure 3 sensors-20-06596-f003:**
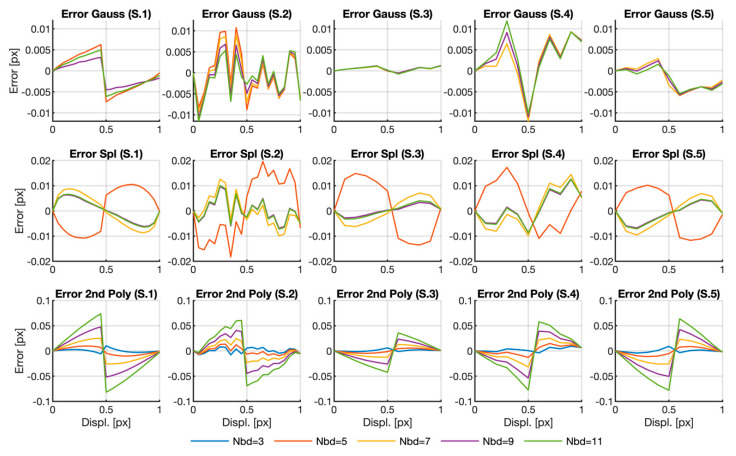
Location error curves obtained for $r^{b}=2$. Curves for the parameter Nbd=3×3 in Gaussian and spline cases have not been represented for better visualization purposes. Curves for the parameter Nbd=5×5 have also not been represented for Samples 3 to 5 in the Gaussian case. The complete graph in Matlab format can be downloaded as a Appendix A from [29].

**Figure 4 sensors-20-06596-f004:**
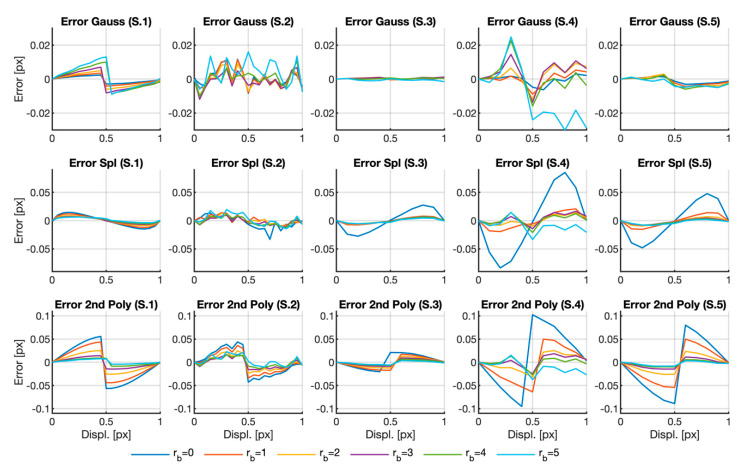
Location error curves obtained for Nbd=7×7. The complete graph in Matlab format can be downloaded as a Appendix A from [29].

**Figure 5 sensors-20-06596-f005:**
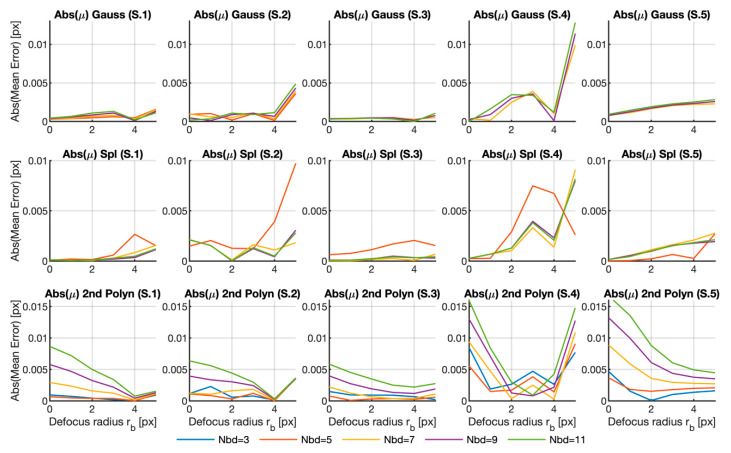
Mean errors calculated for all the samples, fitting functions and different neighborhood sizes versus the Gaussian filter radius. The graphs for the 3×3 region in the Gaussian and spline fitting surfaces were deleted for better visualization purposes. The graphs for the 5×5 region were also deleted for Samples 3 to 5 in the Gaussian case. The complete graph in Matlab format can be downloaded as a Appendix A from [29].

**Figure 6 sensors-20-06596-f006:**
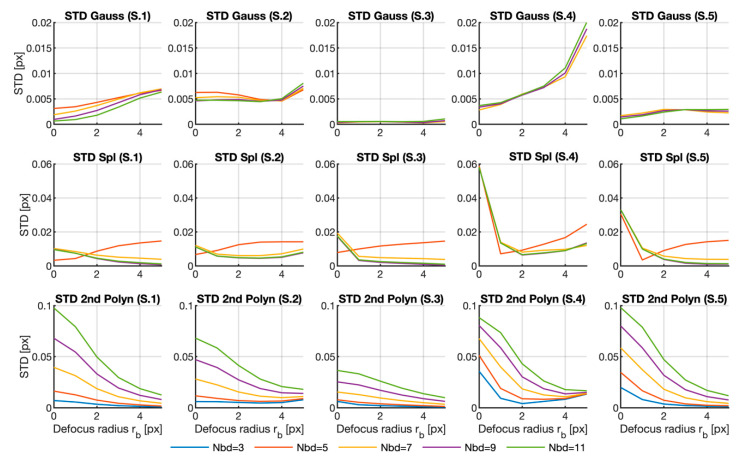
Standard deviations calculated for all the samples, fitting functions and different neighborhood sizes versus the Gaussian filter radius. The graphs for the 3×3 region in the Gaussian and spline fitting surfaces were deleted for better visualization purposes. The graphs for the 5×5 region were also deleted for Samples 3 to 5 in the Gaussian case. The complete graph in Matlab format can be downloaded as a Appendix A from [29].

**Figure 7 sensors-20-06596-f007:**
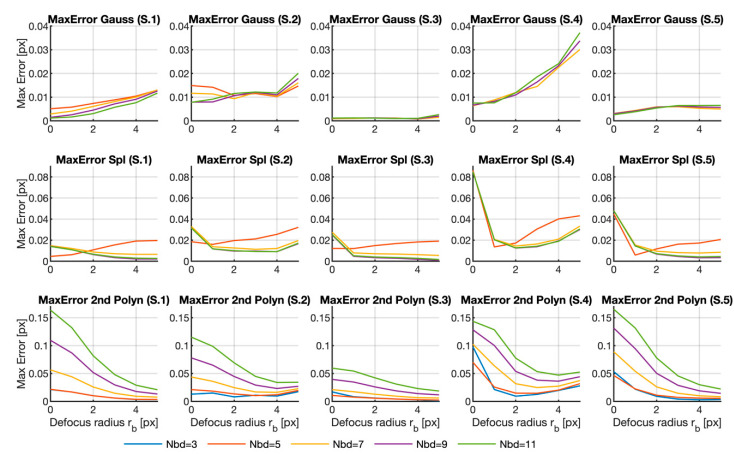
Maximum error calculated for all the samples, fitting functions and different neighborhood sizes versus the Gaussian filter radius. The graphs for the 3×3 region in the Gaussian and spline fitting surfaces were deleted for better visualization purposes. The graphs for the 5×5 region were also deleted for Samples 3 to 5 in the Gaussian case. The complete graph in Matlab format can be downloaded as a Appendix A from [29].

**Figure 8 sensors-20-06596-f008:**
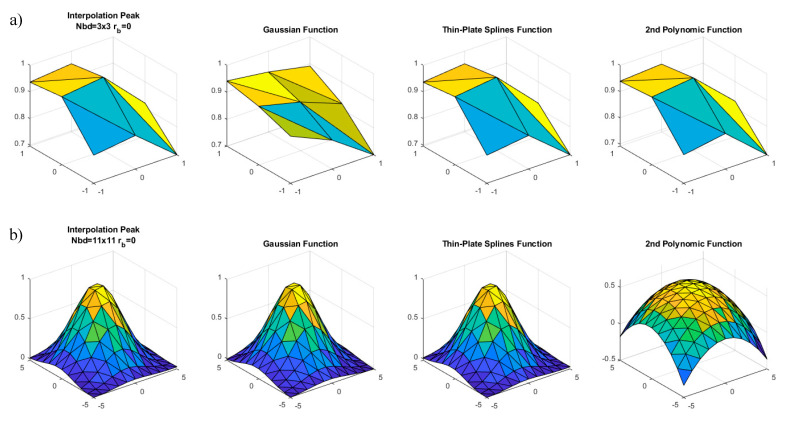
Peak adjustment for Sample 1 with a 0.4 shift using no blur and (**a**) 3×3 and (**b**) 11×11 neighborhoods. For each neighborhood, we show the correlation peak and the reconstructed surface by employing the fitting functions specified above each plot.

**Figure 9 sensors-20-06596-f009:**
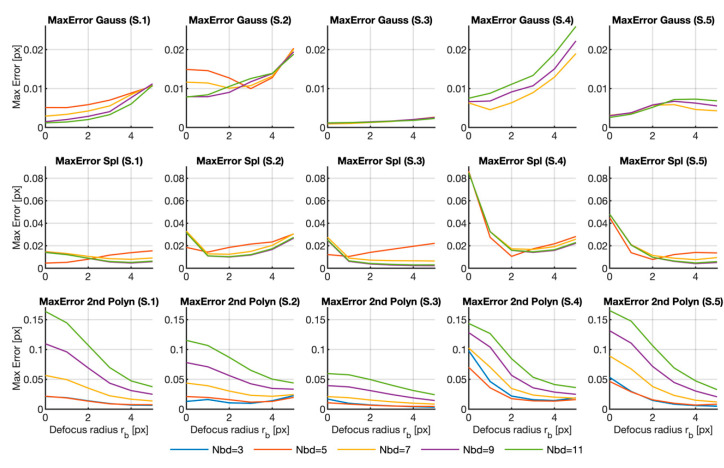
Maximum error calculated when comparing sharp references with blurred templates for all the samples, fitting methods and different neighborhood sizes versus the Gaussian filter radius. The graphs for the 3×3 region in the Gaussian and spline fitting surfaces were deleted for better visualization purposes. The graphs for the 5×5 region were also deleted for Samples 3 to 5 in the Gaussian case. The complete graph in Matlab format can be downloaded as a Appendix A from [29].

**Table 1 sensors-20-06596-t001:** Minimum (best adjustment) and maximum (worst adjustment) values in pixels of μ, STD and maximum error (MaxErr) in each adjustment type with the values of Nbd and $r_{b}$ that provided the results. Below the sample number, the shift between successive frames is specified. The best and worst adjustments in the table are highlighted in red, and both cases were produced with the Gaussian function, with the best in Sample 3 and the worst in Sample 2.

		Gaussian Fit		Thin-Plate Splines		2^nd^-Order Polynomial Fit	
		Best	Worst	Best	Worst	Best	Worst
	µ±σ	4 × 10^−4^ ± 6 × 10^−4^	0.7 ± 0.7	−0.001 ± 7 × 10^−4^	0.004 ± 0.09	−9 × 10^−4^ ± 9 × 10^−4^	0.009 ± 0.1
Sample	MaxErr	0.0012	2.3436	0.0021	0.1105	0.0033	0.1638
1	Nbd	11×11	3×3	9×9	3×3	3×3	11×11
0.05px	r_b_	0	3	5	0	5	0
	µ±σ	9 × 10^−4^ ± 0.005	1 ± 0.7	0.001 ± 0.004	0.006 ± 0.09	8 × 10^−4^ ± 0.005	0.006 ± 0.07
Sample	MaxErr	0.0122	1.9149	0.0096	0.1287	0.0111	0.1152
2	Nbd	11×11	3×3	9×9	3×3	3×3	11×11
0.05px	r_b_	3	4	5	0	5	0
	µ±σ	−3 × 10^−4^ ± 3 × 10^−4^	0.8 ± 0.7	3 × 10^−4^ ± 4 × 10^−4^	−0.008 ± 0.09	2 × 10^−4^ ± 7 × 10^−4^	−0.006 ± 0.04
Sample	MaxErr	7.67 × 10^−4^	2.3761	0.0010	0.1155	0.0012	0.0596
3	Nbd	7×7	3×3	9×9	3×3	3×3	11×11
0.1px	r_b_	4	2	5	0	5	0
	µ±σ	4 × 10^−4^ ± 0.003	0.7 ± 0.6	−0.001 ± 0.006	−0.006 ± 0.1	−0.003 ± 0.004	−0.02 ± 0.09
Sample	MaxErr	0.0063	2.0000	0.0125	0.1537	0.0094	0.1436
4	Nbd	7×7	3×3	9×9	3×3	3×3	11×11
0.1px	r_b_	0	5	2	0	2	0
	µ±σ	9 × 10^−4^ ± 0.001	0.7 ± 0.7	0.002 ± 0.001	0.008 ± 0.1	0.002 ± 0.001	0.02 ± 0.1
Sample	MaxErr	0.0026	1.9000	0.0033	0.1273	0.0034	0.1651
4	Nbd	11×11	3×3	9×9	3×3	3×3	11×11
0.1px	r_b_	0	4	4	0	5	0

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
