# Peer review of "Influence of Neighborhood Size and Cross-Correlation Peak-Fitting Method on Location Accuracyâ€"

_sensors, 2020, doi:10.3390/s20226596_

Round 1
Reviewer 1 Report
The main goal of the article is to compare different functions and preprocessing techniques for the subpixel object/pattern tracking. Authors selected three methods of acurate peak localization, and two meta parameters for them (window size and blur radius). Presented results give some insight into parameter impact, but (as even the authors state) it is hard to tell the one method to rule them all. For different input data different methods may give the best results. My main issue is that in the article only a subset of one dataset was used, and the data was artificially generated. As the authors state (lines 271-272): "(...) this happened because the particular texture of this sequence was not a general rule". It would be beneficial for the article to provide experiment results on the test data with different characteristics.
Authors also state in the introduction and conclusions, that selecting the appropriate function and parameters is key factor. It is easy to select the best function if we have the ground truth data. But how to decide what to choose without this knowledge? Some additional discussion on generalizing the authors findings would be highly appreciated.
The plots of location error (figs 4 to 9) should be presented in vector format, as the very small numbers are hard to read in current form. Also it would be nice if the vertical axes for each method (for each row) have the same scale (it is possible for most of the plots).
Author Response
We acknowledge the reviewer for the comments and agree with the impression that the manuscript is too soft, with many graphs and calculation but without a clear conclusion. After the review, we noticed that we got lots in technical detail, but a main hypothesis and a clear analysis were not clearly written.
The manuscript has undergone a full review and, as a consequence of that, the Title, Abstract, Introduction, Methods and Discussion have been rewritten. Now, we propose an explanation on the peak-locking error from the spatial domain point of view. It is explained as an effect of the excessive weight of the correlation maximum on the local fitting of the peak. Therefore, the fitting method or the neighborhood size play an important role in balancing this effect and, consequently, diminish the location error.
According to this explanation, we think that the results are much clear and also the conclusions. Also, a rule of thumb for speckle images is proposed both in the Abstract and the Conclusions. We hope that now our aim is clearer, and the analysis of the results is less ambiguous.
Regarding the specific details, we have analyzed five sequences from the DIC Challenge platform, that represent a typical pseudospeckle. Samples are public, and our programs and results were also uploaded for the reviewers and will be uploaded to a public platform, so that any researcher can check and use our results and algorithms.
Results here presented compare the influence of different parameters on 5 typical samples of the same kind and many of the error reductions achieved are really subtle, that could be masked by the experimental noise in a real experiment. Therefore we use these database sequences images instead of another ones recorded from real experiments. Additionally, notice that the public images used are specifically designed to test sub-pixel correlation methods, therefore they are very well suited to our goals.
Regarding the specific sentence in lines 270 and following is misfortunate. We meant that the explanation for the specific gap that appeared for rp=4 relies in the sample, since it is not a general rule. In fact, there is a general rule in the beahiour that is now better explained in the Discussion section. In brief, since the Gaussian interpolation function decreases at the border, the importance of the most eccentrical samples is low, and becomes lower as the neighborhood gets larger. Therefore, the errors when using this method are relatively insensitive to the neighborhood size.
According to the suggestion, graphs have been changed so that now the scales are the same in each row. Regarding vectorial format, Figures are now uploaded in PDF so we think the quality will be good enough. However, all of them are provided in Matlab to facilitate their analysis and manipulation.
Reviewer 2 Report
The authors present a study about errors associated with determining the 2D-correlation peak location from random displaced images, a technique known as Digital Image Correlation (DIC). In particular the peak-locking effect is considered. Images are downloaded from the DIC challenge webpage and the analysis is performed in Matlab using built-in routines. The paper is well-written, the results properly presented and the discussed. However, I have a few concerns detailed below that leads back to a general feeling of lack of depth and novelty in the manuscript. To be honest, no real quantitative conclusions are provided that isn't already known in the community.
- The origin of the lock-in error isn't discussed. In general, this type of error is associated with an overtone from the sampling that isn't resolved. Sjodahl (Appl. Opt. 33, 6667-6673, 1994), for example, associated this type of error to an aliasing effect that comes from non-resolved features. Other types of overtones can also be introduced numerically (from interpolation) and natural variations in the fields.
- The main reason defocus helps to lower the lock-in error is that the blurring reduces the spatial frequency content in the images that helps to avoid aliasing. A similar approach would be to lower the numerical aperture of the imaging. The way the authors do it is to introduce blurring after sampling the images, that is after aliasing has already been introduced, which is not a relevant comparison. The assumption made by the authors is hence that the images are sampled correctly. Whether or not this is true for the images used is unclear. Their conclusions about the effect of blurring are therefore questionable.
- Eq. (1) is missing a summation sign in the denominator.
- Experimental data is missing. What is the average feature sizes in the images and how large correlation windows are used?
- The random error, e, is known to scale as e=s*sqrt((1-g)/Ng), where s is the half-width of the correlation peak, g is the correlation value and N the number of uncorrelated contributions to the function. How well does such an equation fit the STD results obtained?
- The discussion referring to Fig. 3 in relation to the peak representation is sound and goes back to a more general question of how to best approximate a correlation peak of unknown shape. Obviously a Gaussian assumes a function that is set by 2 (in the case of symmetry) or 3 parameters while the spline assumes all nodal points to be crossed. The Gaussian hence introduce smoothing while the spline doesn't. The authors are however, a bit vague about the settings used for the spline function so no direct comparison can be made. The polynomial is only expected to be correct very close to the peak location so larger windows can be ignored directly unless very large feature sizes are used. This part, that is central to the paper, could be clearer and the appropriate window size in relation to the actual peak width should be stated.
In conclusion, the paper is well written. However, the overall lack of details and apparent novelty prevents me from recommending publication.
Author Response
We acknowledge the encouraging words from the reviewer about the quality of the manuscript and the constructive comments. After the review, we notice that the main issue of the paper could be better explained, as well as the introduction and the discussion. Therefore, we have reviewed all the manuscript and rewritten many parts of it, including the Abstract, the Methods and the Discussion, that has been split from the Results section.
The main changes are due to the inclusion of an explanation of the peak-locking origin. We knew that the origin of this error is due to a sampling problem and is usually analyzed in the Fourier domain, as a problem of aliasing. In his sense we were not aware of the early works from Sjödahl, which help us very much to clearly formulate our hypothesis.
Although it may seem off-topic, our approach to the problem comes from our previous work with fitting corneal surfaces with Zernike Polynomials. Corneal surfaces are fitted from the local radius of a set of rings projected onto the corneal surface. There, we noticed that, because of the geometry, the outer samples had a heavier influence on the total surface of the central samples, which are scarce but are the most important part of the optical surface. Also, the intersection of the rings with the CMOS matrix imposed some sampling problems that also distorted the results [Espinosa et al. Journal of Modern Optics, 19, pp. 1710 (2019).10.1080/09500340.2011.556263]. By translating the general idea of surface fitting to the correlation problem, we interpret that the peak-locking effect may be due to the lack of samples and the excessive weight of the maximum on the result. Of course, this is a consequence of the inefficient sampling but, up to our knowledge, may be easier to interpret to the people that is not familiar with signal processing.
According to this, the Introduction has been rewritten and extended and the Discussion of the results is done from this perspective. Also, a final recommendation for this kind of images is given in both the Abstract and the Conclusions.
Below, please find the answer to the specific comments:
Comment 1.
The Introduction has been rewritten and the origin of the peak-locking error explained. According to this, the use of Gaussian or quadratic functions is justified. Also the analysis of the results is done from this perspective. New references have also been added to the text recognizing the existing work in the issue.
Comment 2.
From the new perspective, blurring helps to the fitting, but it is redundant with the effect of the Gaussian fitting. In a similar way, double blurring is also redundant and thus, can be omitted from the testing.
In the text, we explicitly say that the filters are applied a posteriori. Of course, a proper experiment maximizing the use of dynamic range and oversampling the texture will help very much, but it is not always possible. Images used are generated artificially, with speckle sizes ranging from 4 to 10 px (the DIC site says that the average size is 5 px). We have checked this out by autocorrelation. In fact, we also checked the homogeneity of the image by testing different areas and with different sizes. According to this we can admit that the image is correctly sampled, accordingly to the Nyquist limit. Of course, when we take a local neighborhood and interpolate the peak, there appears a new problem of correct sampling the correlation peak, which is the one that affects to the error described in the manuscript.
Therefore, according to our proposal and the results obtained we think that our results are according to our hypothesis and thus, correct.
Comment 3: Corrected.
Comment 4
We agree with the reviewer that many details about the calculation and methods were not clear. The Methods section has been revised and many details have been added. We have also added details about the detail size, template size and maximum calculation.
All methods and results have been uploaded to a public repository so that they are disposable for testing.
Comment 5:
We have checked the random error and we obtained similar values for all the samples and blurring radii. For no defocus the values obtained were around 0.007, while from maximum defocus the error obtained was 0.02, which is obvious, since many high frequency details are lost. We did not find any connection between these errors and the error represented in the manuscript, and do not explain the variations observed, so we decided to not include these calculations in the manuscript.
Comment 6:
Figure 3 has been moved to the Discussion section. Although the figure is explained in the text, we still think that may be necessary to understand the magnitude of the effect.
Details about the calculation of the maxima and the algorithms are mentioned in the text. Notice that spline fitting is used here as a middle case between the Gaussian and the polynomic case, so a detailed explanation about the exact method will distort the manuscript.
Please notice that Gaussian is defined by 6 parameters i.e. a1+a2.*exp(-((x-a3)./a4).^2).*exp(-((y-a5)./a6).^2) and also the quadratic case. Therefore, the result does not rely on the number of parameters, but in their shape. Additionally, we are using smoothing spline (thin plate algorithm is just a two-dimensional generalization of a cubic spline) so, in our opinion, the comparison is adequate.
In any case, the programs are included in a public repository, so the algorithm can be easily checked.
Regarding the comment about the main issue of the manuscript, we believe that now with all the changes, the manuscript is better focused and all the details better linked.
We hope that, with all the changes introduced you find our manuscript worth to publish in Sensors.
Reviewer 3 Report
The purpose of this paper is to analyze the effect of different interpolation functions on the peak-locking error. However, this paper didn’t present theoretical models that reveal their relationship.
Only simulations on Matlab software were performed to analyze the peak locking errors of three
traditional interpolation methods by varying some parameters. The authors concluded that "the
selection of the appropriate interpolation function around the relevant peaks is the key point to
reduce the peak locking error" and “the interpolation of the correlation peak is robust”. In my opinion,
these conclusions are obviously and provide little guidance for real measurements. Moreover,
the authors didn’t carry out actual experiments to validate the validity of their points of view.
More research work should be done and hopefully more innovative analytical methods will be proposed.
Author Response
We agree in the fact that the text could be better explained, as can be concluded from the reviewer’s comments. According to them and those from other reviewers, we have rewritten the Abstract, Introduction, Methods and Discussions and we think that now it is more clearly written and our work and findings are better explained.
Now, we propose an explanation on the peak-locking error from the spatial domain point of view. This effect is explained as an effect of the excessive weight of the correlation maximum on the local fitting of the peak. Therefore, the fitting method or the neighborhood size plays an important role in balancing this effect and, consequently, diminishing the location error. The discussion is also re-oriented according to this perspective.
The methods section has also been revised, with a better description of the methods and the samples used. Notice that all the programs used and the results are at free disposal in a public repository.
Also, a rule of thumb for speckle images is proposed both in the Abstract and the Conclusions. We hope that now our aim is clearer, and the analysis of the results is less ambiguous.
Although experiments with non-synthetic images may add some information, we think that the implementation errors would distort the effects described and would add a variability that will not be easily replicable in other experiments by other people. Additionally, the tested images are taken from a bank that is at public disposal for testing these kinds of algorithms and facilitate the replication of the result by any researcher. Therefore, we think that the images used are better than any others that could be obtained through an experiment.
We hope that with all the changes in the manuscript you consider it better explained and worth publishing in Sensors.
Reviewer 4 Report
The content of you paper is interesting but there is a problem with the presentation, i.e. the goal, the structure and the balance.
You did not introduce an original algorithm but your approach based on existing algorithm is interesting. This should be better visible in the title of your paper.
There is no "State of Art" (and the algorithms you use could be part of it!) and the "method" only takes 3 pages ... and the results start at page 5 (within 15 pages).
You should better structure, organize and balance your paper.
Author Response
You should better structure, organize and balance your paper.
We agree with the comments about the general structure of the manuscript. Consequently, the text has been thoroughly revised and many sections heavily changed, including the Abstract.
Title has been changed to give a better first glance of the paper and to reinforce the key points of it.
Now, the Introduction has been extended including, among other issues, the origin of the peak-locking error has been explained. New references have been added in order to update the State of the Art. According to this, the aim of the manuscript has been refined and better explained.
The methods have been also rewritten and the programs and results have been published in a public repository. A Discussion section has been split from the Results section, so the analysis is more comprehensive and precise.
Finally, a final recommendation is proposed as a result of all these changes.
We hope that you now find the manuscript and our results better structured and adequate for publication in Sensors.
Round 2
Reviewer 2 Report
The authors have updated the manuscript or responded convincingly to my original comments. I now believe the manuscript to be of high enough quality to be accepted by sensors.
Author Response
We acknowledge the comments of the reviewer. We are also convinced that the manuscript has really improved with her previous suggestions and the modifications introduced.
Reviewer 3 Report
The authors have revised the title and introduction to explain the motivation of the study more clearly.
From the spatial domain point of view, the effects of the cross-correlation method were analyzed and the main factors affecting the peak-locking errors were clarified.
A more detailed and full discussion for the results of image processing and parameter effects has been presented and clearer conclusions were drawn.
The explanation is reasonable for using DIC images to verify the effectiveness of these methods.
Author Response
We appreciate very much the reviewer's comments. We also hope that, with the included explanation and analysis the manuscript, together with the programs, is useful for the community.
Reviewer 4 Report
You modified some parts of your paper, and you added some material: this is a significant improvement. But you didn't modify the structure of the paper, and this was the MAIN REQUIREMENT ... and I insisted on this point. I really don't understand why you submitted again this paper, not taking this fundamental remark into account!
Modifying the structure of the paper is something easy to do on one hand, because you already have all the scientific material, and difficult to do on the other hand, because you have to put the same material in a different structure.
Let me help you in doing it.
- you have a 2 pages introduction, a 3 pages (or 2 and 1/2) for a section entitled "methods" (what is it?) and the rest of the paper about "results, discussion". This is not correct, and "methods" doesn't mean anything correct in such a context.
- "Introduction" is too long in this context ... but let say it's correct
- "Result and discussion" is correct
- BUT "methods" is not correct at all: it must be split into two sections "state of the art" (in which you describe the different approaches that have been published and on which you base your analysis) and another one in which you describe YOUR CONTRIBUTION; this second part is usually entitled "method" (because it is the method you propose) and not "methods", but you can entitle it in another way (if you wish to) if it shows that it is your contribution. Same thing for "state of the art": you can have another title such as "other works" ...
- "state of the art" and "method" represent the main part of the paper ... and it cannot be only 2 pages and a half on a 17 pages paper (at least, 5 to 6 pages).
Author Response
We are sorry that the reviewer feels that we did not follow his requirements. The manuscript was balanced, from our point of view, by adding a clear hypothesis and a separate section for discussing the results.
The structure of the manuscript is the typical in the field of Physics and Engineering. Moreover, in this journal, the Instructions for Authors clearly state the contents of each section at https://www.mdpi.com/journal/sensors/instructions#manuscript:
"Research Manuscript Sections
- Introduction: The introduction should briefly place the study in a broad context and highlight why it is important. It should define the purpose of the work and its significance, including specific hypotheses being tested. The current state of the research field should be reviewed carefully and key publications cited. Please highlight controversial and diverging hypotheses when necessary. Finally, briefly mention the main aim of the workand highlight the main conclusions. Keep the introduction comprehensible to scientists working outside the topic of the paper.
- Materials and Methods: They should be described with sufficient detail to allow others to replicate and build on published results. New methods and protocols should be described in detail while well-established methods can be briefly described and appropriately cited. Give the name and version of any software used and make clear whether computer code used is available. Include any pre-registration codes.
- Results: Provide a concise and precise description of the experimental results, their interpretation as well as the experimental conclusions that can be drawn.
- Discussion: Authors should discuss the results and how they can be interpreted in perspective of previous studies and of the working hypotheses. The findings and their implications should be discussed in the broadest context possible and limitations of the work highlighted. Future research directions may also be mentioned. This section may be combined with Results.
- Conclusions: This section is not mandatory, but can be added to the manuscript if the discussion is unusually long or complex.
- Patents: This section is not mandatory, but may be added if there are patents resulting from the work reported in this manuscript".
Following this structure, the State of Art is included in the Introduction. Noticed that 22 out of 30 references are cited in this Section. The Materials and Methods section should include all the information that allows replicating our study. This includes the experimental setup, algorithms and approximations used. In our case, there is no experimental setup, so we only included the algorithms and parameters under study: we have explained here the samples used, the method for the analysis, the interpolation functions and their application domain and the effect of the blurring filter. Additionally, a flow chart explaining the algorithm is included. With all of this, a researcher with some programming skills should be able to reproduce our results. In any case, the source code for our algorithms is also included in a public repository. In this sense, we just followed the name suggested without the “materials” word, since we are not using any material or experimental setup. We also find that the plural name is adequate since three different fitting functions are used so, in the end, three different subpixel methods are presented, although the programs have very much in common. In any case, we think that this is a minor grammatical question that does not affect the scientific content of our contribution.
According to this, suggestions from the reviewer are, somehow, against the instructions to authors since these parts should be included in the Introduction.
It is possible that the structure proposed by the reviewer is usual in other areas, but in Physics and Engineering, the State of Art is usually included in the “Introduction” and the “Materials and Methods” include an explanation of the algorithms, and that is what we expect to find when we approach a paper.
In summary, our impression is that the manuscript agrees with the Journal recommendation, so we are sorry but cannot follow the reviewer's recommendations regarding this issue.
However, we think that their suggestions about the balance of the manuscript in the first round where very useful to us and made us to reorganize the manuscript with a clear improvement in the clarity and readability of the manuscript.
We hope that, despite our partial disagreement about the manuscript structure, the reviewer understand our reasons and still finds the manuscript worth for the community and accepts it for its publication in Sensors.